# Economic evaluation protocol of a short, all-oral bedaquiline-containing regimen for the treatment of rifampicin-resistant tuberculosis from the STREAM trial

Laura Rosu ,[1] Jason Madan ,[2] Eve Worrall ,[3] Ewan Tomeny ,[1] Bertel Squire ,[1] on behalf of STREAM Study Health Economic Evaluation Collaborators

¹Centre for Applied Health Research and Delivery & Department of Clinical Sciences, Liverpool School of Tropical Medicine, Liverpool, UK
²Warwick Medical School, University of Warwick, Coventry, UK
³Department of Vector Biology, Liverpool School of Tropical Medicine, Liverpool, UK

**Correspondence to**
Laura Rosu;
laura.rosu@lstmed.ac.uk

## ABSTRACT

**Introduction** A December 2019 WHO rapid communication recommended the use of 9-month all-oral regimens for treating multidrug-resistant tuberculosis (MDR-TB). Besides the clinical benefits, they are thought to be less costly than the injectable-containing regimens, for both the patient and the health system. STREAM is the first randomised controlled trial with an economical evaluation to compare all-oral and injectable-containing 9–11-month MDR-TB treatment regimens.

**Methods and analysis** Health system costs of delivering a 9-month injectable-containing regimen and a 9-month all-oral bedaquiline-containing regimen will be collected in Ethiopia, India, Moldova and Uganda, using 'bottom-up' and 'top-down' costing approaches. Patient costs will be collected using questionnaires that have been developed based on the STOP-TB questionnaire. The primary objective of the study is to estimate the cost utility of the two regimens, from a health system perspective. Secondary objectives include estimating the cost utility from a societal perspective as well as evaluating the cost-effectiveness of the regimens, using both health system and societal perspectives. The effect measure for the cost–utility analysis will be the quality-adjusted life years (QALY), while the effect measure for the cost-effectiveness analysis will be the efficacy outcome from the clinical trial.

**Ethics and dissemination** The study has been evaluated and approved by the Ethics Advisory Group of the *International Union Against Tuberculosis and Lung Disease* and also approved by ethics committees in all participating countries. All participants have provided written informed consent. The results of the economic evaluation will be published in a peer-reviewed journal.

**Trial registration number** ISRCTN18148631.

## BACKGROUND

The STREAM trial is a phase III non-inferiority randomised controlled trial (RCT) to test the efficacy, safety and economical impact of shortened multidrug-resistant tuberculosis (MDR-TB) treatment regimens. MDR-TB is a form of tuberculosis (TB) caused by bacteria that cannot be treated with two of the most

### Strengths and limitations of this study

► The economic evaluation of STREAM will be the first study to estimate the costs incurred by both patients undergoing multidrug-resistant tuberculosis treatment and the healthcare system within a phase III randomised controlled trial.

► The detailed costing and analysis in four different settings will provide valuable insights into the timings and drivers of the costs associated with implementation of a 9-month all-oral bedaquiline-containing regimen. The study will generate important evidence needed for future policy decisions and the shaping of targeted interventions.

► The trial setting means that additional research costs (e.g. costs for collecting pharmacokinetic samples, social support costs paid for by the study) that would not be incurred in a routine setting will be incurred. These research costs will be separated out and eliminated from the costing analysis. Additionally, the experience of participants and delivery of health services (e.g. frequency of visits) will in places, inevitably deviate from routine practice, with implications for patient and health system costs. Though we will attempt to adjust for these differences in analysis, guaranteeing no interference may not be possible.

powerful, first-line anti-TB drugs, isoniazid and rifampicin. Globally, in 2017, there were a little over half a million people with TB resistant to rifampicin, and out of these, 82% had MDR-TB.[1]

The WHO's End TB Strategy is among the health targets of the Sustainable Development Goals. It was adopted by the World Health Assembly in 2014 with the aim of reducing TB deaths by 90% and new cases by 80% between 2015 and 2030, as well as reducing to zero the number of households incurring catastrophic costs due to TB by 2020. Currently, global TB incidence is falling at 2% per year, which is

insufficient to reach the 2020 milestone.[2] This means that new ways of addressing the disease must be found to meet these targets. Careful evaluation of alternative treatment strategies is vital to ensure the most effective and feasible approaches are implemented.

The December 2019 WHO rapid communication recommends the use of shorter, all-oral, bedaquiline-containing regimens for patients with MDR-TB.[3] It seems that all-oral regimens, as opposed to those containing injectables, are becoming the preferred option for treatment of MDR-TB as data from the South African TB programme had suggested them to improve patient outcomes. Replacing the injectable with bedaquiline resulted in better treatment success and better adherence.[3] Besides the clinical benefits, it is also thought that the all-oral treatment leads to lower costs from a health system and patient perspective.[4] It is therefore crucial to test these hypotheses via an RCT in multiple settings. Furthermore, to date, no phase III trial has included an economic analysis of the 9-month bedaquiline-containing regimen, making it difficult for policymakers to assess the economical and financial impact. STREAM is the first randomised phase III trial to include such an analysis, to compare the all-oral, bedaquiline-containing and injectable-containing 9–11-month MDR-TB treatment regimens.

### Objectives

The questions that the economical evaluation is aiming to address include:

► What are the health system costs of treating patients with MDR-TB using the following regimens: a 9-month injectable regimen; a 9-month all-oral bedaquiline-containing regimen and a 6-month injectable regimen?
► What costs do patients face during and after treatment?
► How does MDR-TB affect patients' socioeconomic situations?
► What financial coping mechanisms do patients employ?

The primary economical objective is to estimate the cost utility of the two MDR-TB interventions, in each country, from a health system perspective. To achieve this, an economical evaluation of both the costs and consequences associated with each intervention will be conducted.

Secondary economical objectives include assessing the cost utility of the regimens from a societal perspective and evaluating the cost-effectiveness of the regimens from both a health system and societal perspective.

The effect measure for the cost–utility analysis will be the QALY, while the effect measure of the cost-effectiveness analysis will be the efficacy outcome from the clinical trial that is favourable or unfavourable.

## METHODS AND ANALYSIS

### Randomised controlled trial design

Health economics data will be collected alongside the STREAM trial. Its protocol has been published elsewhere.[5] In brief, the STREAM study is an international, multicentre, parallel-group RCT of patients with MDR-TB and patients with rifampicin-resistant and isoniazid-sensitive TB. It will be assessed whether the proportion of participants on regimen C with a favourable efficacy outcome at week 76 is not less on that on regimen B, that is, C is non-inferior to B. Data will also be collected on regimen D for secondary comparisons. Treatments administered are outlined in figure 1 and explained below. Trial recruitment started in April 2016, across 13 sites in 7 countries (table 1).

At the start of Stage 2, randomisation was to regimen A, regimen B, regimen C and regimen D, in a ratio of 1:2:2:2, done using a web-based system managed by Medical Research Council Clinical Trials Unit (MRC CTU). Version 8.0 of the protocol limits randomisation to arms B and C, so patients will no longer be randomised to regimen A and regimen D and randomisation will be in a ratio of 1:1. At least 200 patients to each of regimen B and regimen C will be randomised, across all sites. This was determined based on the assumption that the proportion of patients with a favourable efficacy outcome at week 76 is 80% for regimen B and 82% for regimen C. With a non-inferiority margin of 10% and a one-sided significance level of 2.5%, 180 evaluable patients will be required in each of the two regimens to demonstrate non-inferiority.

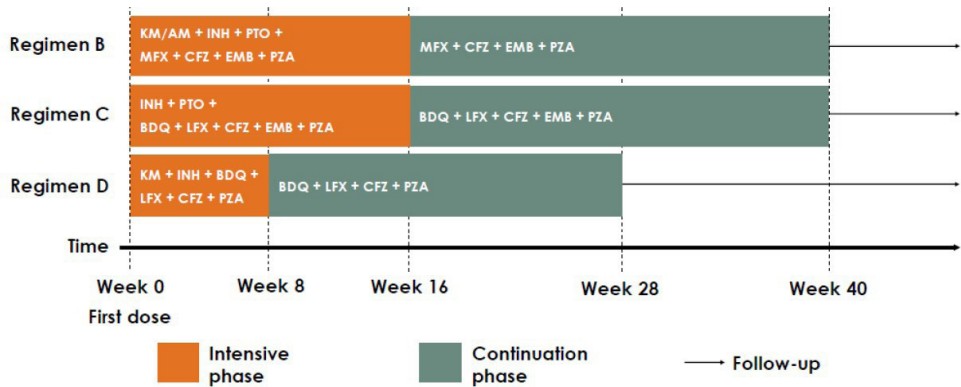

**Figure 1** Treatments outline. Regimen A was dropped of the trial.

| Table 1 | STREAM trial sites | |
|---|---|---|
| | **Clinical trial sites** | **HE sites** |
| Mongolia | National Center for Communicable Diseases, Ulaanbaatar | |
| Ethiopia | Armauer Hansen Research Institute, Addis Ababa | x |
| | St. Peter's Hospital, Addis Ababa | x |
| South Africa | King Dinuzulu Hospital, Durban | |
| | Helen Joseph Hospital, Johannesburg | |
| | Empilweni TB Hospital, Port Elizabeth | |
| | Doris Goodwin, Pietermaritzburg | |
| Moldova | IMSP, Chiril Draganiuc, Chisinau | x |
| Uganda | Mulago Hospital, Kampala | x |
| Georgia | National Center for Tuberculosis and Lung Disease, Tbilisi | |
| India | B.J. Medical College, Ahmedabad | x |
| | National Institute for Research in Tuberculosis, Chennai | x |
| | Rajan Babu Institute for Pulmonary Medicine and Tuberculosis, Delhi | x |

If 10% of patients will be excluded from the primary efficacy analysis population, a total of 400 patients would be required in total for regimens B and C[5].

The health economic analysis will include participants of the clinical trial in the above-mentioned sites, who are over 18 years old and fulfil the inclusion/exclusion criteria as outlined in the trial protocol. All patients in the study will be followed up until week 132, with the primary analysis conducted on data collected up to week 76.

Patient data will be collected at 12-week intervals, during the patient assessment visits for the clinical trial, using a questionnaire developed based on the STOP-TB questionnaire, in all health economic sites.

Health system cost data will be collected by the focal health economists in each country during the whole trial period.

The Consolidated Health Economic Evaluation Reporting Standards checklist has been used as a guide to optimise the preparation and reporting of the methods used (online supplemental annex 1).

### Health system resource use and costs

A mixture of top-down and bottom-up approaches will be used.

Data regarding staff time and staff activities involved in the management of MDR-TB treatment for each regimen will be collected by the focal health economists in each country using a standardised questionnaire developed by the health economic team, pilot tested in all HE sites and used in the first phase of the trial.[6]

A full assessment of the health system costs of delivering the MDR-TB regimens, including tests performed, consumables used, inpatient stay costs, drugs administered and overheads, will be done in each country, for each arm. Any relevant resource events will also be included. These will be collected by the focal health economists in each country using hospitals' accounting records, clinical trial casa report forms (CRFs) and STREAM protocol, and will be costed using local unit costs where possible. Where this will not be possible, STREAM or in-country private healthcare facilities unit costs will be used.

The costs associated with the diagnosis and management of serious adverse events caused by MDR-TB or its treatment will also be included. The costing will include all tests performed, examinations, investigations, inpatient stays and medication received, as well as staff costs. Data will be collected in an event costing tool developed in Microsoft Excel by the HE trial team and the main data source will be the clinical trial CRFs.

The total health system costs for each trial arm will be estimated by summing the costs of each resource used and presented by the following cost elements, by phase (see table 2).

Capital costs extending beyond 1 year (eg, equipment) will be annualised over their expected lifespan assuming a discount rate of 3%.

Research costs such as costs related to the pharmacokinetics study will not be collected or included in this economic evaluation. The health system costing will be done in close collaboration with the central health economic team to make sure it is sensible and evaluated with the support of a team of clinicians involved in the clinical trial. If deemed appropriate, other research costs that do not reflect usual practice will be excluded.

### Patient costs

Patient costs will be collected by administering questionnaires that have been developed based on the STOP-TB questionnaire.[7] Data will be collected in two stages. First, a baseline questionnaire will capture socioeconomic data of each patient before they start treatment. Then, a follow-up questionnaire capturing any changes to the socioeconomic data and a patient treatment cost questionnaire will be administered every 12 weeks.

The patient costs to be collected are presented in table 3.

The total direct cost per participant receiving MDR-TB treatment will be calculated as follows:

$$\text{Total direct cost} = (\text{CostDots} * \text{NoVisitsD}) + (\text{CostSVisits} * \text{NoVisitsS}) + (\text{CostUVisit} * \text{NoVisitsU}) + \text{CostSupp}$$

where NoVisitsD, NoVisitsS, NoVisitsU=number of visits for attending DOTs, scheduled and unscheduled visits, respectively.

Usually, patients with TB are accompanied by a guardian to the direct observed treatment (DOT) and/or assessment visits. The guardians' direct costs (transport, food and accommodation costs) for each patient and for each visit will be included in the patient–costs analysis. Patients

**Table 2** Health system costs sources and calculation methods

| Cost element | Unit | Data sources | | Method |
| | | Costs sources | Quantity used per treatment phase (intensive, continuation and follow-up until week 76) | |
| --- | --- | --- | --- | --- |
| Inpatient stay | Cost per day | Local hospitals' accounting records or local private facilities if not available | Actual number of inpatient stay days for all patients | Unit cost per day multiplied by the number of inpatient days for each patient |
| Laboratory tests | Cost per test | Local hospitals' laboratories or local private facilities if not available | Frequency from the STREAM trial protocol | Cost per test multiplied by the number of tests performed for each patient |
| Medication | Unit cost per tablet/dose | Local hospitals' pharmacies purchasing lists (alternative drug price lists if not available locally) | Dosages, treatment interruptions, etc, from the STREAM trial clinical CRFs | Unit cost per dose multiplied by the total number of doses for each patient |
| Staff | Cost per minute | Local pay scales | Time collected using staff questionnaire | Unit cost per minute multiplied by number of minutes in a visit multiplied by number of total visits |
| Social support | Cost per week | TB national programme | TB national programme | Cost per week times number of weeks the patient is eligible for social support |
| Consumables | Per patient per visit | Local hospitals' pharmacies purchasing lists or local private pharmacies | Quantity of each unit collected via direct observation and staff questionnaire | Unit cost per patient per visit multiplied by the number of visits. |
| Serious adverse events (SAEs) | Per patient per SAE | A combination of all the above | A combination of all the above | Unit costs of: consumables, lab tests, medication, staff will multiplied by the quantity of each to calculate the cost of managing each SAE |
| Overheads | Overhead costs per patient per day | As reported by the local hospitals accounting records | As reported by the local hospitals. Number of patients in the TB unit will be used as a proxy. | Total overhead costs will be calculated for the TB unit over a year, then divided by the number of patients with TB in a year |

who indicate they had a 'guardian' during treatment will be asked whether this guardian lost an income when accompanying them; their lost time will be assumed to equal the patient's and valued at the national minimum wage.

All participants, conditional on survival to week 76, will be included in the primary analysis. In the secondary analysis, all modified intention to treat participants will be included, treating missing answers as missing data and handled as explained in the missing data section below.

All costs will be collected in the local currency and converted to US$ using the exchange rate reported by OANDA[8] at the time of the analysis. All costs will be inflated to 2021 prices.

Due to logistics issues, data collection for the health economic component was delayed at two Indian sites, Ahmedabad and Chennai, and the Ugandan site, so baseline and week 12 patient data will be collected at the week 24 or week 36 visit for the first patients enrolled

into the trial. This will be subject to sensitivity analysis. All interviews after week 36 will be conducted as scheduled, during the patient assessment visits.

The analysis will be performed in Stata (Stata, USA) and for each cost category, descriptive statistics (mean, median, SE and IQR) will be presented.

Quality assurance exercises will be carried out regularly during data collection by the central Health Economics team, to assess the logic and credibility of responses. Feedback will be provided to data collection staff on any issues raised from the exercise, so that they could correct and improve their guidance to participants during data collection.

### Health-realted Quality ofLife measurement

For the primary outcome calculations, patient health states will be measured prospectively using the EQ-5D-5L[9] every 12 weeks from week 0 (i.e. baseline), before the patient takes the first drug, until week 76. The responses

**Table 3** Patient cost data collection method and analysis plan

| Cost type | Data collection method | Analysis |
|---|---|---|
| Cost of attending direct observed treatment (DOTs) (CostDots) | Through patient CRFs (transport and food costs data) | For each cost type category, data will be aggregated for each site and arm, to estimate the mean direct cost per visit |
| Costs of attending injection DOTs (CostDots) | Through patient CRFs (transport and food costs data) | |
| Patient cost for attending scheduled patient assessment visits (CostSVisits) | Through patient CRFs (transport and food costs data) | |
| Patient costs for attending unscheduled patient assessment visits (CostUVisits) | Through patient CRFs (transport and food costs data) | |
| Food supplements (CostSupp) | Through patient CRFs | Mean spend for each time point to be calculated and presented as the cumulative difference in food purchases between arms |
| Income loss during and after treatment | Reported by patients if willing to reveal their income at each time point; if not, working hours reported to be used as a proxy | If patients are unwilling to reveal their income, average salary values from the specific areas in each country will be used. The total lost hours will be multiplied with the hourly average wage. Total income loss during treatment and follow-up will be calculated |

to the questionnaire will be converted into health utility scores using the most appropriate tariff for each country, selected based on geographical proximity and economical context. Currently, the tariffs that we propose to use are from Indonesia (for India), Ethiopia (for Ethiopia and Uganda) and Poland (for Moldova) and can be seen in online supplemental annex 2. We will use updated value sets if these become available before the analysis stage. The value sets will be used to calculate the HRQoL for each patient at each interview point. Observations for each patient will be combined to calculate a QALY score for each arm using the 'area under the curve' linear method, using the formula below:

$$QALY = \sum \left[ \frac{(U_i + U_{i+1})}{2} \right] \times (t_{i+1} - t_i)$$

where $U$=utility value and $t$=time between interviews.

QALY calculations will also account for mortality during the follow-up period, by assigning 0 QALYs from time of death until the end of follow-up.

The health system costs will be calculated on a per patient basis and together with the QALY outcome will be used to calculate the incremental cost-effectiveness ratio (ICER) of regimen C to regimen B, using the formula below:

$$ICER = \frac{(Cost_{RegimenC} - Cost_{RegimenB})}{(Mean\ QALY_{RegimenC} - Mean\ QALY_{RegimenB})}$$

Cost-effectiveness acceptability curves will be constructed to compare the regimens' probabilities of being cost-effective against a set of pre-set threshold values, ranging from US$0 to US$100 000 and including some published estimates.[10]

## Secondary objectives

Secondary objectives will consider the primary clinical outcome in the clinical trial. This is a favourable outcome,

where a participant had their last two culture results, taken on separate visits but no more than 6 weeks earlier than week 76, negative or an unfavourable outcome.

For the societal perspective analyses, direct patient costs data collected as explained above will be added to the health system costs to calculate the societal costs.

## Subgroup analyses

We will present data disaggregated by age, sex, HIV status, site and other variables may be presented where they will be identified in the study as potentially relevant.

## Missing data

The nature and pattern of missing data will be analysed. If necessary, multiple imputation techniques[11] will be used to address the missing data in the base case, by using relevant baseline variables. This method is recommended for economical evaluations alongside clinical trials.[12] Other methods such as complete case analysis, average imputation, lowest and highest point imputation and listwise deletion will be tested in the sensitivity analysis.

## Statistical analysis

We will present our results in terms of precision, that is, how close the data are expected to be to the true population value, presenting means and SD of the results. 95% CI ranges will be constructed and presented such that there is a 95% probability that the results will contain the true population parameter.[13]

## Sensitivity analyses

Sensitivity analyses will be used to test the robustness of the results. Planned sensitivity analyses can be seen in table 4 ; however, any other things that become important will also be tested.

A non-parametric bootstrapping approach will be used to determine the level of sampling uncertainty

**Table 4** Planned sensitivity analyses

| Parameters | Rationale/method |
|---|---|
| Complete-case analysis, Average imputation, lowest and highest point imputation | If the level of missing observations for costs and HRQoL is higher than 10%, the MI technique is more prone to bias. Data sets will be analysed to assess whether the results indicate similar conclusions |
| Patient data collected retrospectively in India and Uganda | As some data have been collected retrospectively during the trial due to logistics issues, two data sets, one including the retrospectively collected data (where recall bias might have occurred) and one excluding it, will be analysed to assess whether the results indicate similar conclusions. |
| On the most important cost drivers | Unit costs will vary across different sites in the same country. Therefore, deterministic sensitivity analysis will be conducted to assess whether the results change as unit costs of the most important cost drivers are varied within plausible ranges. |
| Parameter uncertainty | Probabilistic sensitivity analysis to explore uncertainties surrounding key parameters; 1000 simulations will be run, and results presented as mean costs and QALYs. |
| Inpatient stay | Since 2011, WHO recommends outpatient models of care for patients with multidrug-resistant tuberculosis. The analysis will be re-run excluding inpatient stay costs |

surrounding the mean ICER by generating 1000 estimates of incremental costs and outcomes. These will be presented on a cost-effectiveness plane. CIs of the generated ICERs will then be calculated, in order to summarise the uncertainty due to sampling variations.

Net monetary benefit (NMB) will be calculated for each bootstrap estimate for a range of cost-effectiveness thresholds as follows:

$$NMB = (\lambda * QALYs) - Costs$$

where $\lambda$ represents the cost-effectiveness threshold. This will be calculated as one to three times Gross Domestic Product (GDP) per capita, and other thresholds from country guidance or the literature. The regimen with NMB>0 or with the highest NMB should be adopted. Mean NMB will be reported with 95% bootstrap CIs and z-test conducted.

### Patient and public involvement

WHO's End TB Strategy includes policy goals around elimination of patient catastrophic costs, and this study has been developed to measure and inform both public and stakeholders regarding the economical impact of MDR-TB on patients.

The health economic research questions were developed based on the STOP-TB questionnaire by the health economic team involved in conducting the study at Liverpool School of Tropical Medicine and University of Warwick, based on clinical practice, trial protocol and literature review. All health economic questionnaires have been pilot tested with opportunity for patients to give feedback.

Community advisory boards (CABs), comprised of volunteers from (among others) community-based organisations, those affected by TB and sometimes trial team members, are functioning with the support of the trial at all 13 STREAM Stage 2 sites. Most CABs were formed at site initiation and, therefore, did not inform the development of the research question and outcome measures; however, input on the trial protocol was received from the Global TB CAB. The STREAM CABs act as coordinating mechanisms for community engagement at STREAM trial sites. Their activities include community outreach (engaging the local communities and key populations to raise awareness and literacy on MDR-TB, research, and the trial), provision of psychosocial support to study patients and advocacy activities aimed at improving programmes and policies. The CABs also meet regularly with their respective study teams for trial updates and to pass on patient and community feedback from the trial. Results of the trial will be disseminated to participants and affected communities, with the support of STREAM CABs, likely at outreach events for participants and their families.

The burden of the intervention will be assessed by the patients taking part in the health economic component of the trial, through the EQ-5D-5L questionnaire, which is a self-reported measure of quality of life. These patients will also assess the economic impact the disease had, by reporting changes in income and employment status throughout the study.

### COVID-19 impact

Also, the COVID-19 outbreak started during the trial. Lockdown has been imposed on 18th March in Uganda and on 24th March in India, while Moldova and Ethiopia declared state of emergency in March 2020. It is expected that the COVID-19 mitigating measures taken in most countries will affect the socioeconomic status of the patients and their quality of life, independent of their MDR-TB or MDR-TB treatment.[14] There are a few measures that will be taken to record this. A COVID-19 diary, containing information about the lockdown

restrictions, will be completed by each site (see online supplemental annex 3). Also, an additional questionnaire has been developed to further explore some of the answers regarding their income, spending and health-related quality of life.

As data collection started in 2016, before the outbreak, the lockdown imposed will be modelled as an independent explanatory variable for parameters such as quality of life, working hours and supplements spending during intensive, continuation and post-treatment phase. If the variable turns out to be significant, we will use it to adjust values reported post pandemic, using model predictions of what would have been reported if the pandemic hadn't happened.

Additional changes to the protocol as a result of COVID-19 may be implemented as needed.

## DISCUSSION

STREAM will be the first study to estimate the costs incurred by both patients undergoing MDR-TB treatment and the healthcare system within a phase III RCT.

The detailed costing and analysis in four different settings will provide insights into the timing and drivers of the cost saving or dissaving of implementing a 9-month all-oral bedaquiline-containing regimen, providing the data for targeted interventions if needed.

The study will have certain limitations. The EQ-5D-5L is not a condition-specific measure, and so may miss differences in symptoms that are important to participants. Also, our method assumes a linear relationship between values at different time points; however, this might not be accurate. It was considered not feasible to ask participants to complete the EQ-5D-5L questionnaire at a more frequent interval, that is, each DOT visit.

The trial setting also means that the experience of participants might be different from routine practice, in ways that could influence costs, such as the frequency of visits and their location and the provision of support (eg, transport vouchers, food vouchers).

**Correction notice** This article has been corrected since it first published. The provenance and peer review statement has been included.

**Collaborators** STREAM Study Health Economic Evaluation Collaborators: Mamo Girma, Vanita Patel, Makwana Mukesh, Malaisamy Muniyandi, Shravan Kumar, Sangeetha Subramani, Saleem Ahmad, Jasper Nidoi, Irina Pirlog, Mariana Macarie, I.D. Rusen, Gay Bronson, Meera Gurumurthy, Karen Sanders, Sarah Meredith, Andrew Nunn, Ben Spittle, Wendy Dodds, Robyn Henry-Cockles, Rachel Bennett, Elisa Giallongo, Danni Maas, Rachel Bennett, Ruth Goodall, Saiam Ahmed, Claire Cook, Katharine Bellenger, Gopalan Narendran, Bruce Kirenga, Elena Tudor, Rajesh Solanki, Daniel Meressa, Adamu Bayissa, Anuj Bhatnagar, STREAM community advisory boards (CABs).

**Contributors** SBS has obtained research funding, is the principal investigator of the study and contributed to the original design of this economic study. LR drafted the protocol and contributed to the design of this study. JM, EW and ET provided helpful feedback for all aspects of the work, contributed to the design of the study and revised the draft manuscript.

**Funding** This article is made possible by the generous support of the American people through the United States Agency for International Development (USAID). The contents are the responsibility of the authors and do not necessarily reflect the views of USAID or the United States Government. Stage 2 is funded by the USAID and Janssen Pharmaceuticals, with additional funding from the UK Medical Research Council and the UK Department for International Development (DFID), under the USAID Cooperative Agreement GHN-A-00-08-000040-00. It is sponsored by Vital Strategies, Inc. (an affiliate of The International Union Against Tuberculosis and Lung Disease).

**Competing interests** None declared.

**Patient consent for publication** Not required.

**Provenance and peer review** Not commissioned; externally peer reviewed.

**ORCID iDs**
Laura Rosu http://orcid.org/0000-0003-1172-0962
Jason Madan http://orcid.org/0000-0003-4316-1480
Eve Worrall http://orcid.org/0000-0001-9147-3388
Ewan Tomeny http://orcid.org/0000-0003-4547-2389
Bertel Squire http://orcid.org/0000-0001-7173-9038

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
