## [Reviewer comments · BMJ Open]

ARTICLE DETAILS

TITLE (PROVISIONAL)	Economic evaluation protocol of a short, all-oral bedaquiline-containing regimen for the treatment of rifampicin-resistant tuberculosis from the STREAM trial
AUTHORS	Rosu, Laura; Madan, Jason; Worrall, E; Tomeny, Ewan; Squire, Bertie

VERSION 1 – REVIEW

REVIEWER	Abela Mpobela Agnarson Johnson & Johnson Global Public Health, New Brunswick, NJ, USA I'm employed by Johnson & Johnson , whose subsidiary, Janssen, is the manufacturer of bedaquiline
REVIEW RETURNED	28-Aug-2020

GENERAL COMMENTS	Submission and publication of research protocol aim to promote evidence-based practice. Thank you for giving me the opportunity to support your work. The reviewed study is relevant in the discussion on the treatment of MDR-TB and resource utilization. The structure of the manuscript is complete. The aim and method are concise, with alignment with CHEERS Checklist. The figures (figures are in low resolution and difficult to read) and tables are relevant. It is a well-written manuscript. Specific Comments Abstract • L52: I suggest to replace the term “a nine-month all oral regimen” with “nine months alloral bedaquiline-containing regimen” to reflect the title of the publication.• L53: Include the fact that the data from India and Uganda are to be collected retrospectively. Strengths and limitations of this study L32: Replace “an all-oral regimen” with a “nine months all-oral bedaquiline-containing regimen” and throughout the manuscript. Same applies to objectives section, p.6/L34 L37: The author state that due to the study being collected from multiple settings, resulting in numerous experiences by participants. The author will adjust for the differences in research costs. It is not clear here if the author refers to the charges to conduct the study or the costs recurring for patients and the health system during the study period?
---

	Background L14-15: I am wondering if the author could include the reference to the claim? L19: There study evaluating the cost-effectiveness of all oral nine-month bedaquiline containing regimen. Moreover, the WHO consolidated guidelines on tuberculosis Drugresistant tuberculosis treatment Module 4: Treatment Online annexes: Page 125-127, includes analysis on all oral nine months bedaquiline-containing regimen. The uniqueness of the suggested study is that it is the first multi centre economic evaluation study from a randomized phase III trial. L34: Replace the term “a 9-month non-injectable regimen” with “nine months all-oral bedaquiline-containing regimen”? Methods and Analysis General comment: The authors outline a clear and concise process of data collection for STREAM study from a clinical trial perspective. I would recommend the authors also highlight the data collection process for HE data considering that the data collection for Uganda and India has already been collected. Secondly, it would be informative if the authors could include a questionnaire that was used in Uganda and India in the current publication. Randomised controlled trial design Page 7, L16: The authors assume inclusion of a total of 200 participants in the study. For efficacy, the assessment will be based on 200 patients per arm (for each of the arms B, C), but in the objectives of the economic assessment, arm D (6 month injectable regimen) is mentioned too. It is recognized that some countries will have lower recruitment rates, and some participants may not choose to participate in the economic evaluation study. What is the accepted number of participants from each site? How are the authors accounting for potential uncertainty in the estimates? Health system resource use and costs Table 2:  • Include the follow-up timeline • Extend the table, including a description data sources Patient costs The authors intend to measure the cost drivers behind the illness and treatment of TB. I strongly suggest to the authors to include the costs components incurred cost of guardians/accompanying persons and lost time HRQoL measurement It is not clear if the study will assess HRQoL scores at baseline? Also, how is the percentage change in score over the treatment period will be compared? Page 9-L58: I suggest to author to include references to existing tools. Page 10-L22: Please provide a reference to the threshold values here.
--	---

	Page 10:L32: What specific patient data parameters? Include an example, please. Page 10:L52: Do you refer to means and standard deviations? If so, please state that. Page 11: L58: Provided reference to the claim, also include the time points for Covid-19. I also suggest to include the questionnaire if you have already developed one. Page 12: L8: The recruitment for the trial started in 2016. Please specify what sort of data was collected before 2015 that you anticipate to include.
--	---

REVIEWER	Ronnie Matambo Biomedical Research and Training Institute, Harare, Zimbabwe
REVIEW RETURNED	04-Sep-2020

GENERAL COMMENTS	Dear Authors This writeup clearly shows a lot of effort and attention has been put into this work. The study objectives, study design including the methods are well articulated. Best wishes.
---

VERSION 1 – AUTHOR RESPONSE

The authors wish to thank the reviewers for expending the time to review our manuscript and for the insightful comments. The authors have made every effort to comply with the comments received. Specific responses are provided below each comment and a revised manuscript with tracked changes submitted.

Specific Comments

Abstract

- L52: I suggest to replace the term “a nine-month all oral regimen” with “nine months all-oral bedaquiline-containing regimen” to reflect the title of the publication.

Response: Replaced here and throughout the document

- L53: Include the fact that the data from India and Uganda are to be collected retrospectively.

Response: We have made it clearer in Table 4 P10 that we are referring to the patient cost data that will be collected retrospectively in India and Uganda. We have also explained under the patient costs sub-heading (P8) what data will be collected retrospectively. The newly added paragraph reads: “Due to logistics issues, data collection for the health economic component was delayed at to Indian sites, Ahmedabad and Chennai, and the Ugandan site, so baseline and week 12 patient data will be collected at the week 24 or week 36 visit for the first patients enrolled into the trial. This will be subject to sensitivity analysis. All interviews after week36 will be conducted as scheduled, during the patient assessment visits.” Having made these changes in the text we prefer to maintain the abstract as it stands.

Strengths and limitations of this study

L32: Replace “an all-oral regimen” with a “nine months all-oral bedaquiline-containing regimen” and throughout the manuscript.

Same applies to objectives section, p.6/L34

Response: This has been replaced here and throughout the document

L37: The author state that due to the study being collected from multiple settings, resulting in numerous experiences by participants. The author will adjust for the differences in research costs. It is not clear here if the author refers to the charges to conduct the study or the costs recurring for patients and the health system during the study period?

Response: We have provided clarity, showing that we are referring to the research costs in the HE patient and health system analysis. The new sentence now reads: The trial setting means that additional research costs (e.g. costs for collecting pharmacokinetic samples, social support costs paid for by the study) that would not be incurred in a routine setting will be incurred. These research costs will be separated out and eliminated from the costing analysis. Additionally, the experience of participants and delivery of health services (e.g. frequency of visits) will in places, inevitably deviate from routine practice, with implications for patient and health system costs. Though we will attempt to adjust for these differences in analysis, guaranteeing no interference may not be possible.

Background

L14-15: I am wondering if the author could include the reference to the claim?

Response: WHO reference included (reference number 3).

L19: There study evaluating the cost-effectiveness of all oral nine-month bedaquiline containing regimen. Moreover, the WHO consolidated guidelines on tuberculosis Drug-resistant tuberculosis treatment Module 4: Treatment Online annexes: Page 125-127, includes analysis on all oral nine months bedaquiline-containing regimen.

The uniqueness of the suggested study is that it is the first multi centre economic evaluation study from a randomized phase III trial.

Response: Sentence has been amended as follows: Furthermore, to date, no phase III trial has included an economic analysis of the nine-month bedaquiline-containing regimen , making it difficult for policymakers to assess the economic and financial impact. STREAM is the first randomised phase III trial to include such an analysis, to compare the all-oral, bedaquiline-containing and injectable-containing 9-11 month MDR-TB treatment regimens.

L34: Replace the term “a 9-month non-injectable regimen” with “nine months all-oral bedaquiline-containing regimen”?

Response: This has been replaced here and throughout the document

Methods and Analysis

General comment:

The authors outline a clear and concise process of data collection for STREAM study from a clinical trial perspective. I would recommend the authors also highlight the data collection process for HE data considering that the data collection for Uganda and India has already been collected.

Secondly, it would be informative if the authors could include a questionnaire that was used in Uganda and India in the current publication.

Response: A new paragraph has been added under the patient costs sub-heading to include details on the data collected retrospectively (P8). Also, two new paragraphs have been added under the Randomised controlled trial design section (P5): Patient data will be collected at 12-week intervals, during the patient assessment visits for the clinical trial, using a questionnaire developed based on the STOP-TB questionnaire, in all HE sites.

Health system cost data will be collected by the focal health economists in each country during the whole trial period.

We understand the point regarding the availability of the questionnaire; however, the details of this questionnaire cannot be shared at this time as the trial is ongoing and this instrument is still in use at all sites. We plan to share the questionnaire in a future paper when data analysis has been completed.

Randomised controlled trial design

Page 7, L16: The authors assume inclusion of a total of 200 participants in the study. For efficacy, the assessment will be based on 200 patients per arm (for each of the arms B, C), but in the objectives of the economic assessment, arm D (6 month injectable regimen) is mentioned too. It is recognized that some countries will have lower recruitment rates, and some participants may not choose to participate in the economic evaluation study. What is the accepted number of participants from each site? How are the authors accounting for potential uncertainty in the estimates? Response: Thank you for highlighting this point. All participants, except one, enrolled into the trial at the HE sites have given consent to participate in the HE component. As mentioned on page 4, 'Randomised controlled trial design' section in the manuscript, the sample size was determined by the main trial. This is something we have thought about and decided to present the results in terms of precision as explained under the statistical analysis sub-heading. In terms of characterising uncertainty: confidence intervals for the ICER will be calculated, in order to summarise the uncertainty due to sampling variation. Non-parametric bootstrap will be used, based on sampling with replacement from the original data. Confidence intervals will then be constructed using the empirical estimate of the sampling. . This has been made clearer in the manuscript, under the sensitivity analyses subheading.

Health system resource use and costs

Table 2:

- Include the follow-up timeline

Response: Follow-up timeline included- it will be until week 76

- Extend the table, including a description data sources

Response: Table 2 has been extended with a description of data sources included. Additional information on the quantities and the costing methods used have also been added.

Patient costs

The authors intend to measure the cost drivers behind the illness and treatment of TB. I strongly suggest to the authors to include the costs components incurred cost of guardians/accompanying persons and lost time

Response: Thanks for the suggestion. Direct costs incurred by guardians/DOT supporters/accompanying persons will be included in the analysis. A new paragraph has been added under the patient costs sub-heading: Usually, TB patients are accompanied by a guardian to the direct observed treatment (DOT) and/or assessment visits. For each patient with a guardian, their guardian's direct costs (transport, food and accommodation costs) will be included in the patient costs analysis. Patients who indicate they had a 'guardian' during treatment will be asked whether this

guardian lost an income when accompanying them; their lost time will be assumed to equal the patient's and valued at the national minimum wage.

HRQoL measurement

It is not clear if the study will assess HRQoL scores at baseline?

Response: The first sentence under this subheading has been amended to make it clear that HRQoL will be collected at baseline; For the primary outcome calculations, patient health states will be measured prospectively using the EQ-5D-5L⁹ every 12 weeks from week 0 (i.e. baseline), before the patient takes the first drug, until week 76.

Also, how is the percentage change in score over the treatment period will be compared?

Response: Our primary analysis will be the standard one (to calculate and compare QALYs using the area under the curve approach), so we will not be comparing percentage changes in scores, as this has no clear interpretation for cost-utility.

Page 9-L58: I suggest to author to include references to existing tools.

Response: Reference to Appendix 2 has been included in text. Appendix 2 contains the tariff values and their sources.

Page 10-L22: Please provide a reference to the threshold values here.

Response: This has been made clearer in the manuscript: The line now reads: Cost-effectiveness acceptability curves (CEAC) will be constructed to compare the regimens' probabilities of being cost-effective against a set of pre-set threshold values, ranging from US\$0 to US\$100,000 and including some published estimates. The published estimates we are referring to have been referenced.

Page 10:L32: What specific patient data parameters? Include an example, please.

Response: The sentence has been made clearer under the secondary objectives sub-heading: For the societal perspective analyses, direct patient costs data collected as explained above will be added to the health system costs to calculate the societal costs.

Page 10:L52: Do you refer to means and standard deviations? If so, please state that.

Response: Thank you for the suggestion. The text has now been edited, making it clear that we are referring to means and standard deviations.

Page 11: L58: Provided reference to the claim, also include the time points for Covid-19. I also suggest to include the questionnaire if you have already developed one.

Response: The reference has been added. Lockdown/state of emergency dates have been added and COVID19 diary has been included as annex 3. The patient COVID19 questionnaire has not yet been ethically approved, and so has not been included.

Page 12: L8: The recruitment for the trial started in 2016. Please specify what sort of data was collected before 2015 that you anticipate to include.

Response: Thank you for drawing our attention to this. The year 2015 was a typo and has now been corrected. No data collected before 2016 will be included in this study.

VERSION 2 – REVIEW

REVIEWER	Abela Mpobela Agnarson Johnson&Johnson, Sweden I am employed by Johnson & Johnson, whose subsidiary, Janssen, is the manufacturer of bedaquiline.
REVIEW RETURNED	03-Dec-2020
GENERAL COMMENTS	The authors have revised the manuscript in accordance with initial recommendations. I have no further comments.